# Parents and GPs' understandings and beliefs about food allergy testing in children with eczema: qualitative interview study within the Trial of Eczema allergy Screening Tests (TEST) feasibility trial

Clare Clement [1], Matthew J Ridd [2], Kirsty Roberts,[2] Miriam Santer,[3] Robert Boyle,[4,5] Ingrid Muller,[3] Anna Gilbertson,[2] Elizabeth Angier,[3] Lucy Selman,[1,2] Alison R G Shaw[2]

► Prepublication history and supplemental material for this paper are available online. To view these files, please visit the journal online (http://dx.doi.org/10.1136/bmjopen-2020-041229).

For numbered affiliations see end of article.

**Correspondence to**
Ms Clare Clement;
c.clement@bristol.ac.uk

## ABSTRACT

**Aim** To explore parent and general practitioner (GP) understanding and beliefs about food allergy testing for children with eczema.

**Design and setting** Qualitative interview study in UK primary care within the Trial of Eczema allergy Screening Tests feasibility trial.

**Participants** Semi-structured interviews with parents of children with eczema taking part in the feasibility study and GPs at practices hosting the study.

**Results** 21 parents and 11 GPs were interviewed. Parents discussed a range of potential causes for eczema, including a role for food allergy. They believed allergy testing to be beneficial as it could potentially identify a cure or help reduce symptoms and they found negative tests reassuring, suggesting to them that no dietary changes were needed. GPs reported limited experience and uncertainty regarding food allergy in children with eczema. While some GPs believed referral for allergy testing could be appropriate, most were unclear about its utility. They thought it should be reserved for children with severe eczema or complex problems but wanted more information to advise parents and help guide decision making.

**Conclusions** Parents' motivations for allergy testing are driven by the desire to improve their child's condition and exclude food allergy as a possible cause of symptoms. GPs are uncertain about the role of allergy testing and want more information about its usefulness to support parents and help inform decision making.

**Trial registration number** ISRCTN15397185.

## INTRODUCTION

Eczema (synonyms atopic eczema/dermatitis) is a common and burdensome condition, especially among pre-school age children.[1] Clinical guidelines emphasise the importance of avoiding environmental irritants and practising good skin care through

## Strengths and limitations of this study

► We believe this is the first qualitative study to specifically explore general practitioners' (GPs') and parents' views regarding the role of food allergy in children with eczema.

► We interviewed GPs and parents with a range of characteristics and employed a topic guide flexibly to ensure that different aspects of food allergy and testing in children with eczema were captured.

► By virtue of taking part in the trial, to some extent all participants were open to the idea of children with eczema undergoing Skin Prick Tests for food allergies.

► Other healthcare professionals and less well-educated parents may have different experiences and opinions.

regular use of emollients and appropriate use of topical corticosteroids.[2]

A concern among parents of children with eczema, voiced to general practitioners (GPs) and commonly seen in online forums, is the role of food allergy.[3–5] Despite weak evidence to support dietary modification, many parents try excluding foods from their child's diet to reduce eczema symptoms or the need for treatment with medications.[6 7]

In the UK, the National Institute for Health and Care Excellence (NICE) guidance[2 8] recommends that healthcare professionals consider food allergies as potential triggers in children with eczema if they develop symptoms immediately after ingesting a potential allergen, or in those with moderate to severe eczema who have not responded to optimum management.

Immediate-type food allergies are more common in children with eczema[9 10] and food allergy testing may potentially prevent serious allergic reactions and/or identify foods causing eczema symptoms. However, in primary care professional advice as to the importance of allergy testing for children with eczema is variable, as is access to allergy testing.[11] In principle, if allergy testing were shown to inform eczema care then it could be routinely offered in primary care. However, the effectiveness of food allergy testing and associated dietary measures for managing eczema is uncertain.

In addition, there is also uncertainty about the feasibility and acceptability of conducting research to answer the question of if, or when, food allergy testing should be routinely offered to children with eczema. It is important to resolve these uncertainties to help inform parents and GPs decision making about optimising diet and management of eczema.

Nested within the feasibility Trial of Eczema allergy Screening Tests (TEST) Study,[12] we report on a qualitative study that explored parent and GP understanding and beliefs about food allergy testing for children with eczema.

## METHOD
### Study design
The TEST Study was conducted to determine the feasibility of conducting a trial comparing test-guided dietary management versus usual care, for the management of eczema in children. More detail can be found elsewhere,[12] but in brief it was a single-centre, two-group, individually randomised, feasibility randomised controlled trial conducted in 17 GP surgeries in the West of England. Children aged between >3 months and <5 years with mild or worse eczema were randomised to either control (usual care) or intervention. The intervention comprised a structured allergy history and Skin Prick Tests (SPTs) for cow's milk, hen's egg, wheat, peanut, cashew and codfish. Dietary advice was given, based on test results, to continue eating/introduce as normal or to try excluding and reintroducing one or more foods from the child's diet. Where appropriate, referral was made for an oral food challenge, where the child was exposed to a potential allergen under supervision,[12] or to a local allergy clinical for review. All participants were followed up for 6 months.

Semi-structured interviews were conducted with a sample of parents of children in the TEST trial and GPs from participating practices. The aim was to explore participants' beliefs about the role of food allergy in children with eczema, the acceptability of testing and potential barriers to and facilitators of the uptake of food allergy testing in primary care.

### Sampling and recruitment
Purposive sampling was used to capture maximum variation in views and experiences. Parents were sampled from both the intervention and usual care groups and

**Table 1** Topics explored in parent and GP interviews

| Parent interview | GP interviews |
| --- | --- |
| Beliefs about food allergy and their origin | Beliefs about food allergy testing |
| Perceived or experienced acceptability of allergy investigations, including skin prick tests | Views of the acceptability of allergy tests to parents |
| Facilitators of and barriers to uptake of skin prick tests and dietary advice | Facilitators of and barriers to uptake of allergy investigations (including blood and skin prick tests) and dietary advice in primary care |
| Worry or social difficulties related to food allergies | |
| Strategies used to manage their child's eczema, for example, excluding foods | |

GP, general practitioner.

reflected mild/moderate (<17) versus severe (≥17) Patient Orientated Eczema Measure symptom score[13] and socioeconomic status (assessed using the Index of Multiple Deprivation Decile (IMDD) (categories: high 8–10/medium 5–7/low 1–4).[14] Participants allocated to the intervention group were also sampled based on whether they had a negative or positive SPT result. The sampling of GPs captured variation in IMDD[14] of the practice postcode and doctor characteristics (length of time practising as a GP and self-reported confidence in managing eczema (scale 1–10, 1=low, 10=high). Sample size was informed by the team's judgement that we had enough 'information power' to meet the study aims.[15]

### Data collection
Interviews were conducted face-to-face and via telephone by CC (17 parents, 11 GPs), an experienced social science researcher and KR (4 parents), the trial manager with experience of qualitative research. Written consent was obtained for the face-to-face interviews and verbal consent was recorded for the telephone interviews. Interviews lasted between 16 and 43 min (mean 25 min). No notable differences in length or depth of data were seen between face-to-face (6 parents) and telephone interviews (15 parents, 11 GPs). A flexible, semi-structured topic guide was used to assist questioning but allow participants to introduce and discuss new issues (table 1). The full topic guide is available as online supplemental material.

### Data analysis
Interviews were audio-recorded, transcribed verbatim, anonymised and imported to NVivo V.10[16] for data management and coding. Analysis started shortly after data collection started and analytical insights were fed back into further data collection and analysis. Transcripts were analysed thematically using both inductive and deductive

**Table 2** Parent participant characteristics (n=21)

| Parent characteristics | Number of participants |
|---|---|
| Trial arm allocation | |
| Intervention | 15 |
| Comparator | 6 |
| Child's POEM Score | |
| Mild/moderate (<17) | 16 |
| Severe (>17) | 5 |
| Area deprivation score* | |
| Low | 5 |
| Medium | 8 |
| High | 8 |
| SPT results | |
| Negative | 14 |
| Positive | 1 |
| N/A comparator | 6 |
| Education level | |
| Degree or higher | 13 |
| Diploma | 1 |
| A-level | 3 |
| G.C.S.E | 2 |
| NVQ | 2 |

*Index of Multiple Deprivation (14) based on home postcode.
G.C.S.E, General Certificate of Secondary Education; NVQ, National Vocational Qualification; POEM, Patient Orientated Eczema Measure; SPT, Skin Prick Test.

**Table 3** GP participant characteristics (n=11)

| GP characteristics | Number of participants |
|---|---|
| Years' experience | |
| 0–5 | 3 |
| 6–10 | 3 |
| 11–15 | 4 |
| 16–21 | 0 |
| 21+ | 1 |
| Confidence in managing eczema* | |
| Low | 0 |
| Medium | 5 |
| High | 6 |
| Practice deprivation score† | |
| Low | 6 |
| Medium | 1 |
| High | 4 |

*Self-reported scale 1–10, low=1–3, medium=4–7, high=8–10.
†Index of Multiple Deprivation (14).
GP, general practitioner.

codings.[17] Transcripts were coded to establish an initial coding framework and study team members (CC, ARGS, KR and MJR) each independently coded a subset of seven transcripts; any discrepancies were discussed to ensure a coding consensus and maximise rigour.[18] The framework was then applied to all the remaining transcripts by CC. Emergent findings were discussed in regular multidisciplinary trial management group meetings to enhance validity. After coding was completed we drew on the Common-Sense Model to help interpret findings.[19]

### Patient and public involvement (PPI)
Two parents of children with eczema were members of the trial management group and advised on qualitative data collection and analysis. In addition, PPI feedback was incorporated into the final topic guide.

### RESULTS
Twenty-one parents were interviewed from 11 trial practices (table 2). Eleven GPs were interviewed from seven trial practices (table 3). Four main themes emerged from the analysis (two of which related to parent data only): parents' causes and associations, knowledge and awareness, searching for a 'cure' or seeking reassurance around current dietary practice and parents' responses to food allergy test results.

### Parents' causes and associations
Parents seemed unsure of the causes of eczema and discussed several possible factors. They received advice and information from a wide range of sources such as other parents, doctors, family members and the media. Some parents acted on this information, including by removing foods from their child's diet.

Parents believed that family history, age or environmental factors were responsible for their child's eczema, with some discussing how factors could interact or vary depending on the individual.

> I think it must be like a heat thing…it was just to see if something that is genetic that my wife had when she was little and then she grew out of it…But that's really everything we know… But I do have a lot of allergies, so I don't know if she's inherited it. (Parent 1)

> To be honest I don't know an awful lot…I think that obviously dry skin, like water does affect it, but what actually causes it I don't know whether its stuff in the environment…I think there are different causes for different people as well. (Parent 7)

Parents reported being influenced by media coverage of eczema and anecdotal stories from other parents whose child's symptoms improved when certain foods, particularly dairy, were eliminated from their diet.

> I've got a friend that her little girl…was actually allergic to dairy so that resolved some of her skin problems…and so being told by a friend actually dairy's

not good for them and so you prevent them eating certain things…and that will resolve the problem. (Parent 9)

Some parents had been alerted to the possible role of food allergies by speaking with healthcare professionals, including GPs and health visitors, about eczema, who had suggested trying elimination diets or keeping a food diary.

That's what the doctor told us anyway, they said we might need to consider food allergies…they did say that if babies already have allergies, say like a milk allergy, then certain things can be a problem…I think they mentioned soya as well…they said a food diary would be a good idea. (Parent 2)

The health visitor proposed, she suggested that, to run an elimination diet. (Parent 21)

Several parents had explored the role of food in their own child's eczema, before taking part in TEST. In most parent accounts this did not improve the child's condition or improvement was not clearly linked to the change in diet.

It may be food…so we did quite a lot of changing things at the time…we tried keeping a food diary…I think we started off offering a new thing every couple of days just to see if she reacted to it…there was nothing which obviously made her significantly worse…. we thought maybe eggs but then we reintroduced them…and she was fine. (Parent 2)

Whether it was a natural improvement, or it was the milk-free diet, I can't say but that's what happened at that point. (Parent 21)

Some parents, although aware of such stories, were not convinced and wanted more 'evidence' before they acted.

Probably someone has said to me have you tried cutting out dairy and I haven't…If I actually listen to people with their anecdotal things or…my friend's child had eczema and they stopped them having dairy, but I'm a little bit cynical. (Parent 7)

Everyone always says oh there's a link between dairy and eczema, but we could never really find any studies or like have it proven…there's a lot of hearsay but it hasn't actually been proven. (Parent 10)

Some parents believed food allergy may be a factor in their child's eczema by observing a link between their child eating certain foods and their eczema getting worse or eczema occurring when foods were introduced. Others did not see a role or reported having independently 'ruled it out' through exclusion of certain foods from their child's diet.

I used to find if she drank certain things…she would start to scratch and then her eczema started to flare up. Sometimes I would notice after she'd eaten chocolate or something she'd flare up as well. (Parent 3)

I did strongly think it was to do with foods because of when I stopped breastfeeding it was something that flared up. (Parent 12)

I tried cow's milk, all fish, banana…and nothing, no changes…It's not food. I'm sure it's not food. (Parent 17)

### Knowledge and awareness

Knowledge and awareness of food allergy and food allergy testing and its role in managing eczema varied across both parents and GPs. Some parents were aware of what food allergies were before the trial and named common allergens, with some reporting personal experience of such food allergies. However, some parents still reported limited knowledge and understanding of food allergies.

I know there's loads of them. The main ones are dairy and gluten and nuts. That's probably it as far as my knowledge goes. (Parent 17)

I'm not really clued up on much of it…I have certain foods myself my tongue flares up…but other than that I don't really have any clue of food allergies. (Parent 12)

Parents and GPs labelled a food allergy as a set of acute or severe symptoms which could arise from a potential reaction to food. Symptoms which were perceived to be less severe, delayed or gastrointestinal upset were labelled as an intolerance rather than allergy.

So I would say true allergy causes an allergic reaction, so causes an ideated reaction, so usually presents as sort of problems with breathing, lip swelling, skin response so like hives, urticaria, whereas I think intolerance seems to come with GI upset or sometimes can have skin reactions but tends not to create a full blown allergic response. (GP 9)

I think an intolerance [is] just where it might upset them a little bit, you might get mild stomach cramps… whereas an allergy where they mouth might swell up or affect their breathing…more severe. (Parent 11)

Parents who had asked about allergy testing reported frustration with responses from GPs who believed the child did not have severe enough eczema to warrant testing.

I asked if he could have allergy testing just to make sure it wasn't anything like that, but they said they don't tend to do it in young children unless it's [eczema] like severe. (Parent 8)

One parent commented on how they felt that GPs did not have the appropriate information to advise parents:

I don't think there's any solid research to say whether or not there is a link so…it's difficult for doctors to advise on something there isn't any evidence for. (Parent 8)

GPs reported limited experience of and knowledge about food allergy in general and wanted more information to guide their decision making for allergy testing in children with eczema.

> [A] minefield is how I would put it, I think. It's not something we're taught well at medical school…there's a massive online presence about allergy testing, much of it not evidence based. So, there's a big unmet need with parents coming with questions and I think doctors have an unmet educational need. (GP 5)

GPs reported parents frequently requesting food allergy tests for their children. Some more inexperienced GPs reported being uncertain about the evidence and found advising parents difficult. Few of the GPs had referred a child with eczema specifically for food allergy tests, preferring to refer them to a dermatologist for more general advice or telling parents to keep a food diary.

> I don't feel completely confident in knowing its [allergy testing] limitations and uses…we get parents requesting allergy testing and it sort of feels like my training has always suggested there isn't necessarily a role in most cases, but there might be a role in some cases, so I'm not completely clear about the evidence behind it so I find it a bit tricky to advise parents on that. (GP 3)

### Search for a 'cure' or seeking reassurance around current dietary practice

Parents' motivation and willingness for their child to have food allergy testing was influenced by a range of factors. While parents were uncertain of the role of food allergy in eczema, they discussed how food allergy testing might be beneficial in identifying a cause for their child's eczema, providing a cure or helping with management of the condition.

> I just want to get on top of it [eczema] but you just can't because you don't know what is triggering it off…just to rule it [food allergy] out…then I can just stop him from having that food and then it won't bother him. (Parent 5)

They also appeared to seek reassurance or support for their current dietary management strategy; for example, they wanted to know they were doing 'the right thing' for their child by excluding or not excluding foods:

> I'd really like to know if, because I've had so many people say to me about dairy and 'cos I haven't done it, so I'd like to know that I'm doing the right thing. (Parent 7)

However, some parents did express concerns about the impact that identifying a food allergy could have on their family. There were concerns for balanced nutrition and difficulties accommodating different diets within families:

> You need to make sure that they're getting enough fats, particularly for children and protein and fibre and everything else and carbohydrates when you're excluding all of this, and it could be actually quite difficult for some parents, some parents might have other children so the other children are eating one thing and…have to do it for the whole household. (Parent 18)

Some parents were concerned the skin prick test could be uncomfortable for their child and were relieved their child was in the usual care group and did not have it.

> I was quite glad then to be honest he didn't have to do allergy testing…once I saw the fact that he wasn't that happy about being probed and prodded anyway I thought yeah, probably better off not having to do it. (Parent 7)

Some GPs said they believed allergy testing to be appropriate for some children with eczema as it could be useful for informing potential dietary alterations to help manage the eczema. But most GPs, particularly those with more experience, had reservations about the usefulness of testing and were cautious about making referrals to allergy clinics. GPs reported being more likely to refer children for food allergy testing in severe or complex cases and where the cause of the eczema was not clear, and said they were often guided by parental wishes:

> I tend to discourage it [allergy testing] if I'm honest…I think unless we're having problems getting a child's eczema and their symptoms under control…I think in a child who would be very severely affected I would because I think well we're not getting this under control, we need more information…if there was a family history of food intolerance, allergies, those sort of situations. (GP 6)

Whether parents or GPs thought the child may grow out of the eczema also influenced decisions to consider food allergy testing.

> Most children grow out of it and most of the time it's quite mild I think that most of the time people just tend to treat it and not perhaps think about the allergy side of things. (GP 1)

### Parents' responses to food allergy test results

Parents in the intervention group expressed a range of responses to the food allergy test results. Most parents had faith in the healthcare professionals and the test and therefore accepted the results as being accurate.

> I couldn't see any reason not to trust it…I'm pretty trusting in professionals. (Parent 16)

Most results were negative, but parents still found the results useful as they were perceived to either rule out the possibility of food allergy, provide reassurance they were currently acting correctly or confirm what they already

suspected. This gave them a feeling of control over the condition.

> It confirmed what I thought in a way I would have been surprised if she was allergic to something. (Parent 9)

> It was beneficial, so I know now if and when she ever starts eating eggs… we know she's fine with it. (Parent 3)

> It makes a difference. It doesn't change how I treat it but we had a negative result…but it definitely made a difference in terms of ok put your mind to rest… made us a bit more relaxed…And also maybe more feeling of being in control. (Parent 14)

However, some parents had mixed feelings about a negative test result: they were still in the position of uncertainty. They still did not know what causes the eczema or how to manage it, and while some parents were pleased that the test was negative, others were disappointed not to have any answers.

> So, it was mixed emotions, it was like ok that's good but still don't know what's causing it. (Parent 9)

> If it turned out he did have a dairy allergy, we would know that his diet definitely needed to be adapted, whereas I was just guessing most of the time…but then essentially the result of the allergy test was that he wasn't allergic to anything and I was still in the same situation at that point. (Parent 16)

Some parents appeared to accept that their child was 'not allergic' to the foods tested but still had doubts about other foods not tested for.

> I still don't know if he has an allergy to anything that wasn't tested, 'cos they only test for certain ones. (Parent 16)

One parent whose child had received a negative allergy test result still felt that food was a factor. They believed the eczema to be an intolerance to food rather than an allergy, and so the allergy test would not have captured this.

> He didn't have an allergic reaction to milk, they didn't test him on soya…I never thought he was allergic to it, I just assumed he had an intolerance. (Parent 8)

## DISCUSSION
### Summary
We found different uncertainties among parents and GPs regarding the value of food allergy testing in children with eczema. Parents' beliefs around the causes of eczema, including the role of food allergy, and their information sources on this were mixed. Parents expressed few concerns about the limitations of allergy testing, and most were satisfied with the results which gave them a sense of control over their child's condition. Test results

gave them confidence to not change their child's diet but sometimes left them with a desire for more information. GPs felt reluctant to refer for allergy testing due to uncertainty about the effect of testing and dietary management on eczema symptoms.

### Strengths and limitations
As far as we are aware, this is the first qualitative study to specifically explore the views of parents and GPs regarding the role of food allergy in childhood eczema. We interviewed GPs and parents with a range of characteristics and employed a topic guide flexibly to help ensure that all aspects of the role of food allergy and testing in children with eczema were captured. However, all participants were either taking part in (parents) or hosting (GPs) the trial, meaning that to some extent they were all open to the idea of children with eczema undergoing SPTs. We did not explore how age of parents or ethnicity of parents or GPs may influence beliefs about food allergies and eczema and practice. We did not interview GPs from surgeries or parents who declined to take part in the trial. We only interviewed GPs and other healthcare professionals may have different experiences and views. In addition, a high proportion of parent participants were educated to degree level or higher (~60% in the trial); and only one parent interviewed had received a 'positive' test result.

### Comparison with existing literature
Evidence related to parents' food allergy knowledge, attitudes and beliefs is limited, but we know that parents are frustrated by inconsistent or contradictory messages from different doctors,[20–22] and that information online about diet and eczema is readily accessible but often inaccurate or misleading.[22] Our findings are consistent with a recent qualitative synthesis of the eczema literature, which identified a diverse range of beliefs about underlying causes and found that parents sought dietary avoidance as a potential 'cure', removing the need for long-term treatment.[23] Parents in this study were motivated to have food allergy testing to help identify a cure and to ensure they were acting appropriately by including or excluding certain foods. We have reported previously that GPs often either avoid the topic of food allergy in eczema or, if raised, dissuade parents away from testing.[5 24] Our study indicates this may be due to a lack of experience and understanding of food allergy testing. As per Halls *et al*'s[11] analysis of online forums, we identified parents' concerns that dietary restrictions may result in nutritional deficiencies or promote picky eating habits. Our findings suggest the perceived benefits of food allergy testing generally outweigh concerns and lead to parents engaging with food allergy testing.

### Implications for research and practice
Our findings support the need for a definitive trial of test-guided dietary management for childhood eczema. Good quality evidence and resources are needed to guide GPs

on how to advise parents regarding food allergy testing. There needs to be better quantification of where and how commonly parents seek dietary advice for eczema, what changes they make and what the implications for their child and family may be. The views and experiences of a wider range of healthcare professionals, such as paediatric dermatologists or general paediatricians, also needs to be captured.

Meanwhile, until better evidence emerges GPs should continue to follow guidance[2 8] on food allergy testing in children with eczema, specifically seeking specialist advice where it is suspected clinically because of immediate reactions, where there are suggestive symptoms in other organ systems or where the disease is difficult to treat despite optimal topical therapy. Parents are likely to benefit from signposting towards high quality evidence-based information[25] regardless of whether allergy testing is indicated, to help them understand and manage their child's eczema. Alterations to children's diets should be done in conjunction with appropriately trained healthcare professional's advice to avoid unnecessary restrictions.

**Author affiliations**
[1]Bristol Trials Centre, Bristol Medical School, University of Bristol, Bristol, UK
[2]Population Health Sciences, Bristol Medical School, University of Bristol, Bristol, UK
[3]Primary Care and Population Sciences and Medical Education, Faculty of Medicine, University of Southampton, Southampton, UK
[4]Inflammation, Repair, and Development Section, National Heart & Lung Institute, Imperial College London, London, UK
[5]Centre of Evidence-based Dermatology, University of Nottingham, Nottingham, UK

**Acknowledgements** We would like to thank all the GPs and parents that participated in study; West of England Clinical Research Network; and to Catherine Gray, Jo McMeechan and the TEST/BEE Studies PPI group for their contributions. The study was developed with support from UK Dermatology Clinical Trials Network (UK DCTN). The UK DCTN is grateful to the British Association of Dermatologists and the University of Nottingham for financial support of the Network.

**Contributors** MJR conceived the study idea in collaboration with RB, MS and IM; MJR, RB, MS, IM, ARGS and KR developed the initial study design with later input from LS and CC on the nested qualitative study. CC and KR conducted the interviews. CC analysed the data with input from ARGS, LS, MJR and KR. CC prepared the manuscript, MJR, KR, MS, RB, IM, AG, EA, LS and ARGS contributed to drafts of the paper and approved the final draft. CC finalised the paper for submission to the journal. All authors read and approved the final manuscript.

**Funding** This study was funded by National Institute for Health Research (NIHR) School for Primary Care Research (project 383). MJR was funded by an NIHR Post-Doctoral Research Fellowship (PDF-2014-07-013). This study was designed and delivered in collaboration with the Bristol Randomised Trials Collaboration (BRTC), a UKCRC registered clinical trials unit which, as part of the Bristol Trials Centre, is in receipt of National Institute for Health Research CTU support funding. The views expressed in this article are those of the authors and not necessarily those of the NHS, NIHR, or the Department of Health and Social Care.

**Competing interests** MJR: No financial interests; convenes the NIHR SPCR Allergy working group; and was a member of theNational Institute for Health and Care Excellence (NICE) Quality Standard 44 for Atopic eczema in under 12s and RCPCH 'Care pathway for children with eczema' groups. RB has received honoraria for participating in advisory boards for ALK-Abello who manufacture allergy diagnostics and treatments, and DBV technologies and Prota therapeutics who develop food allergy treatments. RB has undertaken expert witness work in legal cases concerning food anaphylaxis or infant formula health claims.

**Patient consent for publication** Not required.

**Ethics approval** The study has been reviewed by the Health Research Authority and given a favourable opinion by the NHS REC (West Midland-South Birmingham Ethics Committee, Reference Number 18/WM/0124).

**Provenance and peer review** Not commissioned; externally peer reviewed.

**Data availability statement** All data relevant to the study are included in the article or uploaded as supplemental information.

**ORCID iDs**
Clare Clement http://orcid.org/0000-0002-5555-433X
Matthew J Ridd http://orcid.org/0000-0002-7954-8823

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
