## [Reviewer comments · BMJ Open]

ARTICLE DETAILS

TITLE (PROVISIONAL)	Parents and GPs' understandings and beliefs about food allergy testing in children with eczema: qualitative interview study within the Trial of Eczema allergy Screening Tests (TEST) feasibility trial
AUTHORS	Clement, Clare; Ridd, Matthew J; Roberts, Kirsty; Santer, Miriam; Boyle, R; Muller, Ingrid; Gilbertson, Anna; Angier, Elizabeth; Selman, Lucy; Shaw, Ali R G

VERSION 1 – REVIEW

REVIEWER	Dr Rebecca Knibb Aston University United Kingdom
REVIEW RETURNED	14-Jul-2020

GENERAL COMMENTS	This is a well written paper exploring parents' and GPs' views, understanding and beliefs about food allergy testing for children with eczema. The rationale for the study is made clear from the introduction. The methods section is quite brief but mainly provides the necessary information required and is clearly written. The results are interesting and provide implications for further research and clinical practice, which are discussed in the discussion section. I have a few minor comments only. In the data collection section only parent interviews are referred to. Could you please provide similar information for the GP interviews? Also, could you say how many were done in person and how many over the telephone? There are four themes developed from the data but only two of them have relevance to the GPs. The results therefore seem more weighted towards parent accounts. Although more interviews were undertaken with parents, it would be useful if the authors could provide some reassurance that GP interview data was fully captured in the two themes presented. There are a couple of instances in the paper where a full reference is provided in the main body of the paper where a reference number is needed.
--

REVIEWER	Ru-Xin Foong Paediatric Allergy Department, Evelina Children's Hospital/Guy's and St Thomas' NHS Trust King's College of London London, UK
REVIEW RETURNED	19-Jul-2020

GENERAL COMMENTS	A very interesting and relevant article in terms of exploration of an area of allergy and eczema that is extremely common in clinical practice but not described in the literature.
---

	Few points to the authors to consider:  - Were there any differences in views based on age of parents or ethnicity of parents/GP - there may be cultural differences in beliefs about food allergies and eczema that might influence practice/referral? - Were the interviews conducted using the information solely in Box 1? In terms of repetition of the study, were more structured interview questions used and is this available for publication as maybe supplementary material? - In the conclusion, you mention signposting parents to high quality evidence based information -- perhaps offer references so if GP's were to read this, they could direct parents to these from the paper? - Consideration for future work - interviewing paediatric dermatologists or general paediatricians who see eczema to see when/what situations make them refer to allergy for testing Minor comments:  - Page 5, line 22 - should be "food allergies" - plural if stating "potential triggers" or food allergy as a potential trigger if singular Page 6 line 25 - tenses change, should be "where the child was exposed to a potential allergen under supervision" Page 6, line 37 - reference (Ridd) not coded numerically - Page 15, Line 25 (reference (Teasdale) not coded numerically instead is written in full within the text)
--	--

VERSION 1 – AUTHOR RESPONSE

Comment number	Comment	Authors response	Page number
Reviewer 1			
1	In the data collection section only parent interviews are referred to. Could you please provide similar information for the GP interviews?	This information has now been added.	5
2	Also, could you say how many were done in person and how many over the telephone?	This information has been added for parents and GPs.	5
3	There are four themes developed from the data but only two of them have relevance to the GPs. The results therefore seem more weighted towards parent accounts. Although more interviews were undertaken with parents, it would be useful if the authors could provide some reassurance that GP interview data was fully captured in the two themes presented.	It is the case that only two of the themes have relevance to the GPs. We now indicate this more clearly.	6
4	There are a couple of instances in the paper where a full reference is provided in the main body of the paper where a reference number is needed.	Full references were provided in the text at the request of BMJ Open editors as the papers had not yet been published. One of	14

		the papers (Teasdale et al.) has now been published and referenced accordingly within the manuscript. The second paper Matt et al. remains under review and has been removed from the manuscript.	
Reviewer 2			
5	Were there any differences in views based on age of parents or ethnicity of parents/GP - there may be cultural differences in beliefs about food allergies and eczema that might influence practice/referral?	Thanks for highlighting this and we agree there might be differences based on these characteristics. Unfortunately, we did not collect this data from interview participants so cannot explore this further in our analysis. We have referred to this limitation on page 13.	13
6	Were the interviews conducted using the information solely in Box 1? In terms of repetition of the study, were more structured interview questions used and is this available for publication as maybe supplementary material?	We have explained how the interviews were semi-structured and guided by a topic which covered the topics in box 1 on page 6. We have also now added the complete topic guide as supplementary material and added a sentence indicating this.	6
7	In the conclusion, you mention signposting parents to high quality evidence based information -- perhaps offer references so if GP's were to read this, they could direct parents to these from the paper?	A reference has now been added.	14
8	Consideration for future work - interviewing paediatric dermatologists or general paediatricians who see eczema to see when/what situations make them refer to allergy for testing	These groups have been added to the sentence recommending future research to capture wider views and experiences of healthcare professionals.	14
9	Page 5, line 22 - should be "food allergies" - plural if stating "potential triggers" or food allergy as a potential trigger if singular	This has been amended.	4
10	Page 6 line 25 - tenses change, should be "where the child was exposed to a potential allergen under supervision"	This has been amended.	5

11	Page 6, line 37 - reference (Ridd) not coded numerically	See reviewer 1 comment 4 response.	
12	Page 15, Line 25 (reference (Teasdale) not coded numerically instead is written in full within the text)	See reviewer 1 comment 4 response.	

VERSION 2 – REVIEW

REVIEWER	Rebecca Knibb Aston University United Kingdom
REVIEW RETURNED	08-Sep-2020

GENERAL COMMENTS	All of my comments have been satisfactorily addressed.
--

REVIEWER	Ru-Xin Foong Paediatric Allergy Department/King's College of London London UK
REVIEW RETURNED	16-Aug-2020

GENERAL COMMENTS	Revisions made from previous comments - no further comments for the authors. I think the paper reads well and covers an interesting topic in the field of food allergy.
---